# Improving Engagement in Antenatal Health Behavior Programs—Experiences of Women Who Did Not Attend a Healthy Lifestyle Telephone Coaching Program

**DOI:** 10.3390/nu15081860

**Published:** 2023-04-12

**Authors:** Jessica Fry, Shelley A. Wilkinson, Jane Willcox, Michaela Henny, Lisa McGuire, Taylor M. Guthrie, Nina Meloncelli, Susan de Jersey

**Affiliations:** 1Dietetics and Foodservices, Royal Brisbane and Women’s Hospital, James Mayne Building Level 2, Butterfield Street, Herston, QLD 4029, Australia; 2School of Human Movement and Nutrition Sciences, The University of Queensland, St Lucia, QLD 4072, Australia; 3Mothers, Babies and Women’s Theme, Mater Research Institute—The University of Queensland, St Lucia, QLD 4072, Australia; 4Impact Obesity, South Melbourne, VIC 3205, Australia; 5Centre for Health Services Research, Faculty of Medicine, The University of Queensland, Herston, QLD 4029, Australia; 6Office of the Chief Allied Health Practitioner, Metro North Health, Royal Brisbane and Women’s Hospital, Herston, QLD 4029, Australia

**Keywords:** pregnancy, gestational weight gain, telephone-counselling, dietitian, antenatal care

## Abstract

Living Well during Pregnancy (LWdP) is a telephone-based antenatal health behavior intervention that has been shown to improve healthy eating behaviors and physical activity levels during pregnancy. However, one-third of eligible, referred women did not engage with or dropped out of the service. This study aimed to explore the experiences and perceptions of women who were referred but did not attend or complete the LWdP program to inform service improvements and adaptations required for scale and spread and improve the delivery of patient-centered antenatal care. Semi-structured telephone interviews were conducted with women who attended ≤2 LWdP appointments after referral. The interviews were thematically analyzed and mapped to the Theoretical Domains Framework and Behavior Change Wheel/COM-B Model to identify the barriers and enablers of program attendance and determine evidence-based interventions needed to improve service engagement and patient-centered antenatal care. Three key themes were identified: (1) the program content not meeting women’s expectations and goals; (2) the need for flexible, multimodal healthcare; and (3) information sharing throughout antenatal care not meeting women’s information needs. Interventions to improve women’s engagement with LWdP and patient-centered antenatal care were categorized as (1) adaptations to LWdP, (2) training and support for program dietitians and antenatal healthcare professionals, and (3) increased promotion of positive health behaviors during pregnancy. Women require flexible and personalized delivery of the LWdP that is aligned with their individual goals and expectations. The use of digital technology has the potential to provide flexible, on-demand access to and engagement with the LWdP program, healthcare professionals, and reliable health information. All healthcare professionals are vital to the promotion of positive health behaviors in pregnancy, with the ongoing training and support necessary to maintain clinician confidence and knowledge of healthy eating, physical activity, and weight gain during pregnancy.

## 1. Introduction

A higher body weight, excess gestational weight gain (GWG), poor diet quality, and low physical activity levels during pregnancy can increase the risk of negative maternal and fetal outcomes, including hyperglycemic and hypertensive disorders, cesarean section delivery, increased labor time, large gestational-age babies, and life-long obesity [1,2,3,4]. Clinical practice guidelines recommend that all women with a pre-pregnancy body mass index (BMI) > 25 kg/m^2^ be referred to a dietitian for health-behavior change interventions [5,6,7,8,9,10] to optimize the health outcomes for both the mother and the offspring.

Despite these recommendations and the previously reported positive impact of well-designed and delivered dietetic interventions on GWG in Australia [11,12], many practice gaps still exist that limit access to and utilization of evidence-based care pathways. These include inadequate funding, a lack of an evaluation of the currently delivered models of care, poor healthcare professional (HCP) knowledge of and referral to the appropriate services, and a lack of HCP confidence and knowledge to discuss weight with women [13,14,15,16]. For women referred to a dietitian for weight management during pregnancy, engagement with and uptake of these services are typically low, particularly for face-to-face services, with 20–50% of women referred not attending appointments [17,18,19,20].

Many health-behavior interventions that aim to improve nutrition quality, physical activity engagement, and/or GWG patterns during pregnancy have been trialed utilizing a range of delivery modes including face-to-face, telephone calls, short messaging services (SMS), smartphone applications, websites, and email [21,22,23,24,25,26]. Despite differences in their intervention content and delivery, these targeted health-behavior interventions show promise for their positive effects on nutrition, physical activity, and GWG. The key components of successful interventions include flexible, multimodal, and interactive delivery, tailored messaging, and the use of multiple behavior-change techniques e.g., goal setting, skill acquisition, and self-monitoring [21,22,23,24,25,26].

In 2018, a quaternary metropolitan facility implemented the Living Well during Pregnancy (LWdP) telephone coaching program to address local referrals and attendance barriers to traditional face-to-face dietetic appointments [27]. Adapted from the Healthy Living after Cancer Program [28], this phone-based intervention targets women with a pre-pregnancy BMI > 25 kg/m^2^ and/or those gaining weight above the Institute of Medicine’s (IOM) recommendations during pregnancy [1]. It aims to support women in achieving GWG within their pre-pregnancy BMI range through behavior change support for healthy eating and regular physical activity aligned with the Australian Dietary Guidelines [29] and physical activity in pregnancy guidelines [30], respectively. This was achieved through information sharing, goal setting, and problem solving, guided by dietitians skilled in motivational interviewing [27]. Participants were supplied with a manual covering a range of topics including goal setting, healthy eating, physical activity, GWG, meal planning, and mindful eating [27]. Throughout their pregnancy, participants were offered up to 10, 60 min telephone calls. Continuity of care was achieved by using the same dietitian throughout the pregnancy and flexible appointments outside of standard work hours. 

Post-implementation evaluation of LWdP demonstrated an increase in dietetic referrals compared to standard care (*n* = 370 vs. 82), improved healthy eating (increased Fat and Fiber Behavior Index Scores 3.03 vs. 3.53, *p* < 0.001), and increased physical activity levels (180 vs. 240 min, *p* = 0.007) for women who completed the program [31]. However, one-third of eligible women declined participation, were non-contactable, or did not attend their first appointment [31]. Of the women who did attend appointments, three out of five did not complete the minimum of four appointments deemed “completion” [31]. Understanding the experiences of women who do not fully engage with nutrition and physical activity antenatal interventions is essential for service improvement and providing patient-centered care. 

This study aimed to explore the experiences and perceptions of women who were referred but did not engage with or complete the LWdP program, to inform program improvements and adaptations required for scale and spread, and improve the delivery of patient-centered antenatal care. 

## 2. Materials and Methods

### 2.1. Study Design

This study employed a prospective, descriptive qualitative approach using semi-structured interviews [32] and was conducted at the Royal Brisbane and Women’s Hospital (RBWH), Queensland, Australia, which provides care for pregnancies across low-risk shared care, continuity of midwifery care, and complex obstetric-led antenatal care. Women who were referred to LWdP between January 2021 and March 2022, and attended ≤2 appointments were invited to participate. The LWdP program has been previously described [31]. In brief, women were eligible for referral to LWdP if they had antenatal care at the RBWH, had a pre-pregnancy BMI > 25 kg/m^2^, and/or were gaining weight above the IOM recommendations [1]. The philosophical approach employed in this study was pragmatic, allowing experiences to highlight what was important to women and valuing the applied over the theoretical [33]. The Consolidated Criteria for Reporting Qualitative Research (COREQ) was used to guide the study design and inform reporting [34]. 

### 2.2. Procedure

Eligible women were identified through clinic records and then contacted via SMS. They were provided with a link to an online registration questionnaire (see Appendix A) on a health-service consumer-oriented survey platform called Citizen Space. The questionnaire also included the study outline, collected demographic information, and asked for consent to be contacted to arrange an interview time. Interviews were conducted with women who had given their consent between February and May 2022 by the lead investigator (JF), who is trained in facilitating qualitative interviews. The interview schedules were designed with clinicians and researchers skilled in consumer co-designed health interventions, translation of research into practice, and/or digital health innovations (see Appendix A). Participants were gifted a $20 grocery voucher for their time. 

Interviews were recorded using the Voice Memos mobile phone application (Apple 2022©). Where interviews could not be recorded, notes were taken by the interviewer. Office365 Microsoft^®^ Word was used to transcribe the audio files and cross-check for accuracy against the audio files by the interviewer. Regular meetings occurred between the interviewer and senior author to debrief on the research processes and the interviewer’s reflections.

### 2.3. Data Analysis

Interview transcripts were thematically analyzed using Applied Thematic Analysis (ATA) [35]. The ATA methodological framework is a rigorous, inductive set of procedures designed to identify and examine themes from data in a way that is transparent and credible [35]. The transcripts were read and re-read to produce the initial codes and later the themes of the interviews. Independent initial coding was undertaken by the interviewer to ensure that this was not unduly influenced by the senior researcher’s values or experiences. The senior researcher (SdJ) then reviewed the transcripts and codes, and additional coding was added where appropriate. A third researcher (SAW) read all the transcripts. The three authors arrived at a consensus on the final themes. There was open reflection across all discussions to ensure that the themes reflected the participants’ rather than the researchers’ views. 

The themes were then mapped against the Theoretical Domains Framework (TDF) [36] and the Behavior Change Wheel (BCW)/COM-B Model of Behavior Change [37]. The TDF and BCW/COM-B selection was informed by Nilsen [38]. The TDF is a determinant framework that was chosen due to its ability to not only categorize barriers and enablers but also to articulate the BCW/COM-B Implementation Theory to enable the evidenced-based selection of interventions to address the identified barriers. All 14 domains of the TDF were considered in the thematic mapping of interviews, with the enablers and barriers sorted into the most appropriate domain/s. The source/s of the behavior were aligned to the relevant COM-B element from the BCW (see Figure 1), potential interventions to support behavior change, and finally, behavior change techniques and how these might be operationalized, drawing from the implementation science literature [37]. 

The terms “nutrition” and “healthy eating”, and “exercise” and “physical activity” were used interchangeably throughout the interview schedules and thematic analysis.

## 3. Results

Of the 228 women referred during the study period, 157 were eligible to participate (see Appendix A for non-responder characteristics). All eligible women were contacted, with ten consenting to be interviewed. Nine interviews were conducted; one woman was uncontactable after consenting. The characteristics of the participants are outlined in Table 1. The average interview time was 16 (range of 9–28) minutes. 

### 3.1. Part 1: Thematic Analysis

Three key themes encompassing seven sub-themes (see Figure 2 and the second column in Table 2) were identified from women’s experiences with referral and exposure to LWdP: (1) the program content not meeting women’s expectations and goals; (2) the need for flexible, multimodal healthcare; and (3) information content and support not meeting information needs. All themes were linked by the overarching theme of women juggling multiple priorities throughout pregnancy.

#### 3.1.1. Theme 1: Program Content Not Meeting Women’s Expectations and Goals


*Initial positive program perceptions*


Overall, positive perceptions of the LWdP program were described by women. Many expressed that LWdP provided information about topics other women wanted to know about and the benefit of obtaining this information from an expert source. 


*“…will help a lot of mums…that like put on weight fast and don’t know why they are…a lot of first-time mums want to know about everything, they want to like search everything up so it’ll be good that they can talk to people… [that] specialize it more than Google”*

*—Participant 6 (Age 22, one appointment attended)*


Women who did not engage with the intervention but received the program manual believed that the LWdP would support mothers in making better choices and having a healthier pregnancy and baby. Women who engaged felt that LWdP was not useful for them but still perceived it would “*be useful to other women*,” (*Participant 8, Age 31, one appointment attended*).


*A highly structured program was delivered but an agile, individualized program was expected*


The program was perceived as being highly structured with guided progression through topics by the dietitians. In contrast, the women who attended a small number of sessions spoke about wanting flexibility with the topics that focused on the health concerns relevant to their desire for knowledge, such as exercise or food safety, or related to a specific stage or symptoms of pregnancy such as hyperemesis. One participant reported a dissonance between the dietitian-directed goals and their own personal goals during a session, which, in part, led to their disengagement in the intervention.


*“…I’ll make my goal…to do with exercise like maintaining, say, three sessions a week or something and then she [dietitian] said oh no this week is meant to be about the healthy eating chapter…”*

*—Participant 8 (Age 31, one appointment attended)*


Another woman described struggling with hyperemesis and wanting meal and recipe ideas to help manage her symptoms and support nutrition intake, which were not provided. 

Women who attended appointments but later dropped out attributed their dropout to their initial expectations not being met. They wanted more specific, tailored, and individualized information and recommendations, including meal plans, specific food recommendations, weight gain targets, pregnancy-safe physical activity recommendations, and more in-depth discussions of program content.


*“…Going more in depth with concepts for people with higher health literacy…”*

*—Participant 3 (Age 35, one appointment attended)*


#### 3.1.2. Theme 2: Need for Flexible, Multimodal Healthcare


*Barriers to appointment attendance*


Multiple women reported finding it hard to commit to and attend all antenatal appointments, including LWdP. They reported struggling to fit appointments around other commitments, including work, childcare responsibilities, and other medical appointments. The structured weekly to fortnightly telephone appointments within the LWdP program were difficult for women to fit into their work schedules.


*“…it’s really hard to work around…I struggled with work, trying to take time off”*

*—Participant 1 (Age 29, one appointment attended)*


Although the LWdP was designed to overcome the barriers to attending face-to-face appointments through its use of telephone consultations, misunderstandings about the telephone nature of the program led one woman to not engage.


*“The timetable [for LWdP] doesn’t suit me…it’s quite far for me to come, especially when I got extra three kids with me”*

*—Participant 9 (Age 35, no appointments attended)*



*Desire for “on-demand” and immediate access to healthcare professionals*


The desire for on-demand information and appointments was expressed by many women due to competing priorities, a reluctance to wait weeks for appointments to ask questions, and some feeling that regular contact was unnecessary. 


*“…[I don’t think] you need to be constantly checked in with every week but you know you can … request appointments or you can have that option of having recurring appointments”*

*—Participant 2 (Age 31, two appointments attended)*


Many women had experienced some form of on-demand healthcare during their pregnancies, with their midwives being readily contactable via SMS 24 h a day. Women found this fit in well with their other commitments while still providing the healthcare they expected.


*“…midwifery program…send a text message … don’t have to wait a week or two weeks to ask a really simple question”*

*—Participant 2 (Age 31, two appointments attended)*



*Need for multiple program delivery modes inclusive of face-to-face and digital*


Digital technology was suggested as a means of engaging with HCPs for receiving trusted source educational material and recording and tracking health goals. A variety of modes were suggested, including interaction with HCPs and the provision of educational resources via SMS, email, websites, and smartphone applications, as well as the tracking of health measurements using digital means, e.g., digital food records. 


*“Apps could definitely assist…you can kind of put in your circumstances and it could…come up with…information or areas of concern for you”*

*—Participant 2 (Age 31, two appointments attended)*


Women were open to using digital health platforms due to their flexibility and convenience for tracking information, receiving educational materials, and contacting their HCPs. 


*“…diary entries of what you are eating…it would be a bit easier having it on the phone ‘cause you have your phone on you all the time”*

*—Participant 2 (Age 31, two appointments attended)*


Conversely, some women suggested that they preferred to see their HCPs either face-to-face or via telehealth rather than a telephone call. 

#### 3.1.3. Theme 3: Information Sharing throughout Antenatal Care Did Not Meet Information Needs


*Different levels of information provided and needed*


Most women received some advice from their HCPs about healthy eating, physical activity, and/or GWG during pregnancy, with midwives reported as the main HCPs providing this information. 


*“They [midwives] asked me to do just like healthy living style…I put [on] a little bit of weight like during the end of the pregnancy and they said that’s normal, but you can just go on the diet—they give me the booklet and I just eat few of the veggies and…healthy foods…”*

*—Participant 9 (Age 35, no appointments attended)*


Women’s satisfaction with the topics covered, the depth of information given, the use of additional resources, and the ongoing discussions throughout their pregnancies varied significantly. For LWdP, women who had received the program manual without interacting with the program dietitian reported that the manual was a satisfactory source of information, whereas women who interacted with the dietitians felt they received no additional information other than what was provided in the manual.


*“I think this booklet have all the information that everyone needed.”*

*—Participant 7 (Age 31, one appointment attended)*


There was a small number of women who reported not being involved in any discussions around healthy eating, exercise, and GWG with HCPs outside of LWdP. Some women reported that they only received information about healthy eating, physical activity, and GWG after they sought it out themselves through internet searches, pregnancy-specific mobile phone applications, and HCP friends.


*“I haven’t really [talked to anyone about healthy eating, exercise or GWG in pregnancy], unless I sought the information out myself…had a look on the internet or I do have a friend who’s a dietitian…”*

*—Participant 8 (Age 31, one appointment attended)*


Minimal information sharing was also described between women and their HCPs about available services to support healthy eating, exercise, and healthy GWG during pregnancy, including the LWdP program.


*“They [midwives] didn’t tell me much about it [LWdP] because they give me the booklet and they say you can go and refer [to] that”*

*—Participant 9 (Age 35, no appointments attended)*



*Assumed knowledge due to multigravida*


Women who had previously been pregnant reported disparities in healthy eating, physical activity, and GWG discussions with their HCPs between their pregnancies. Some attributed this to pregnancy-related complications such as gestational diabetes being their HCPs’ focus, whereas others believed that during later pregnancies they received less education due to having prior “experience”. 


*“I think because this is my second pregnancy, that might like influence the doctor and things”*

*—Participant 8 (Age 31, one appointment attended)*


However, women who did not engage with LWdP with previous gravida reported that they did not need additional support with healthy eating, exercise, or GWG during pregnancy due to their previous experience. 


*“Not really [needing additional information or support], because this not my first pregnancy…”*

*—Participant 5 (Age 32, one appointment attended)*


### 3.2. Part 2. Deductive Mapping to the TDF, COM-B Model, and BCW

The interviews described the barriers to and enablers of engagement with the LWdP program across 9 of the 14 TDF domains (see Table 2, column 3) and 4 of the 6 COM-B Model components (see Table 2, column 4). There was a large overlap across the TDF domains, with barriers and enablers to engagement identified across knowledge, skills (cognitive/interpersonal), beliefs about capabilities, environmental context and resources, and social/professional role and identity. Optimism and beliefs about consequences were identified as only enablers of engagement, whereas social influences were identified as only barriers to engagement. Evidence-based behavior-change techniques (see Table 2, column 6) and potential targeted interventions (see Table 2, column 7) to positively influence engagement with LWdP and antenatal care experiences were identified across six of the nine BCW intervention functions (see Table 2, column 5). These interventions are described in detail in Table 2, column 7 and are broadly categorized as (1) adaptations to the LWdP program, (2) training and support for program dietitians and antenatal HCPs, and (3) increased promotion of health behaviors in pregnancy.

## 4. Discussion

This study explored the experiences and perceptions of women referred to LWdP who did not engage with or complete the program. A lack of individualized content, the program not meeting expectations, and the inability to attend appointments were described as key barriers to initial and continued engagement. Women reported a need for flexible healthcare, with differences reported between women on the ideal approach to care. Differences in the information provided to women during their antenatal care regarding healthy eating, physical activity, and GWG during pregnancy were also evident. These findings have implications for the delivery of person-centered care during pregnancy. Innovative solutions are needed to address the challenges that women face when engaging with health-behavior interventions during pregnancy while working within the funding and delivery constraints of a publicly funded healthcare system. 

Pregnancy is seen as an opportune ‘teachable’ moment for healthy lifestyle changes [39]; however, women lead busy lives with competing priorities, some of which are barriers to engaging with health-behavior interventions [40]. Despite this, women want support for healthy eating, physical activity, and GWG during pregnancy [41]. Expectations of what this support looks like are highly variable, as seen in this study, and range from prescriptive and structured programs to flexible content and contact with HCPs [42]. Study interviews highlighted the need to individualize the content and delivery of LWdP for each woman to meet their expectations, improve motivation to continue the program after the initial session, and provide flexible access to HCPs. The provision of person-centered care through the identification of participants’ expectations of health-behavior interventions, tailoring the intervention content and delivery mode, and simplifying the intervention’s structure to meet participants’ needs and improve participant retention are common [43]. The implementation of pre-program screening to identify women’s goals and expectations of the intervention and preferred delivery mode/s, e.g., in-person, telephone, video call, email, etc., would allow dietitians to tailor content and delivery to the individual, increasing the likelihood of women’s expectations being met and improving engagement and retention. 

The use of digital technology has the potential to provide flexible, multi-modal, person-centered healthcare at a low cost [44]. Women in this study suggested the use of digital technology to provide flexible access to HCPs and health information that can accounting for their busy lives. Digital lifestyle interventions during pregnancy have been shown to be as effective, or more, than traditional care in terms of improving healthy eating, physical activity, and GWG behaviors [25,26], with the tailoring of digital technology health-behavior interventions increasing intervention success [26,44,45]. Multimodal digital technology healthcare interventions have greater success than single-mode interventions, including either multiple digital technology modes [23,46] or digital technology combined with traditional interpersonal healthcare [21,24]. The delivery of the LWdP program via multiple digital technology modes, e.g., email, websites, telehealth, and digital health trackers, including utilizing multiple modes in a single program, could improve women’s ability to engage with the program at the intensity they prefer, with consideration for their existing commitments while retaining the positive intervention outcomes [47,48]. The individualization and adaptation of LWdP requires careful consideration due to the evidence-based nature of the program to ensure consistency of the intervention between women and the maintenance of the evidence-informed behavior-change strategies that are embedded into the content. The maintenance of program fidelity throughout individualization is possible when the program’s core components and theoretical models are maintained [47,48]. For LWdP this would be the delivery of content in ways that supports the underpinnings of the Social Cognitive Theory components of self-efficacy, goal setting, and self-monitoring [49].

Outside of specific interventions, women’s HCPs, especially midwives [50], are in ideal positions to have discussions regarding healthy eating, physical activity, and GWG and initiate referrals to, and encourage engagement with, available support services. As seen in this and other studies [51], women expect their HCPs to discuss healthy eating, physical activity, and GWG with them, especially women who are primigravida. However, many women are confused about the information provided [52], view the lack of discussions on these behaviors by their HCPs as an indication that they are unimportant [53], or are dissatisfied with the inconsistent and brief information provided by their HCPs [52,53,54,55]. When not provided with information, women tend to seek out information from alternative sources [56], which may not always be evidence-based, making HCP-provided advice invaluable and an ideal opportunity to support health-behavior changes. The ongoing support and education of all HCPs working in antenatal care in terms of how to have discussions with women about healthy eating, physical activity, and healthy GWG are needed. Healthcare ‘nudges’, described as subtle changes in environment design or the framing of information to influence behaviors [57], have been shown to improve the promotion of preventative healthcare discussions [58,59]. Increasing nudges within the antenatal care and wider community settings through advertisements promoting and modeling positive health behaviors in pregnancy and available support services could increase the prevalence of healthy eating, physical activity, and GWG discussions; emphasize the importance of and normalize healthy eating, physical activity, and healthy GWG in pregnancy; and increase referrals to and engagement with pregnancy-specific health-behavior interventions.

Although LWdP had initial positive feedback, particularly the program manual, there was some dissatisfaction observed, with dietitians not focusing on individual goals or providing the information women wanted. This dissatisfaction was not investigated or observed in the original service evaluation [31] but likely contributed to dropout. This dissatisfaction may be attributable to the intervention being delivered in ways that did not support women’s personal goals or staff members not having the necessary skills, e.g., motivational interviewing. The abilities of the program facilitator, i.e., appropriate skills and knowledge, and equal partnerships between consumers and HCPs are key to the success of evidence-based interventions [60].

The findings of this study need to be considered in the context of its generalizability limitations. Although the number of women who consented to involvement in the study (6% of eligible) was small, a ‘saturation’ of themes and ideas was achieved. Despite this, the small participant number increased the risk of bias in the study’s outcomes; the views and experiences expressed may not be representative of all women who did not engage with or dropped out of the intervention, especially those from disadvantaged backgrounds, those with lower household incomes, and those with lower education levels. The study’s location, that is, metropolitan Australia, may also limit the global generalizability of the results to antenatal health-behavior interventions. Recall bias may be inherent to experiences expressed by women due to the length of time between the referral to the program and the study interviews. The interviewer’s and primary analysts’ reflexivities may have influenced the themes generated from the interviews. Steps were taken to mitigate this by multiple authors blindly reading interview transcripts before group consensus on the themes was reached. This study is one of very few that explores women’s experiences with healthy eating and physical activity interventions during pregnancy, who did not engage with or dropped out of the interventions. 

## 5. Conclusions

Women require individualized and flexible access to healthcare that can account for their multiple priorities to improve engagement with health-behavior interventions during pregnancy. The information provided should be tailored to meet their expectations for the intervention and support their goals related to healthy eating, physical activity, and healthy GWG during pregnancy. The integration of digital technology via multiple modes into LWdP has the potential to cost-effectively improve access to and engagement with the service. The adaptation of the existing intervention into digital modes requires careful consideration to retain its key evidenced-based components and ensure program efficacy. Local strategies are also needed to address inconsistencies in information provided by HCPs during pregnancy to promote health behaviors and self-management.

## Figures and Tables

**Figure 1 nutrients-15-01860-f001:**
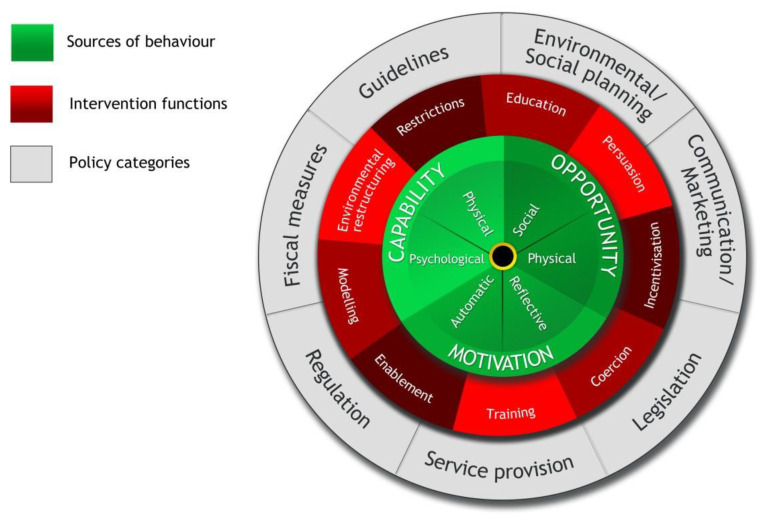
BCW/COM-B Model and their components. Figure by Michie et al. [37]. Utilized under Creative Commons Attribution License.

**Figure 2 nutrients-15-01860-f002:**
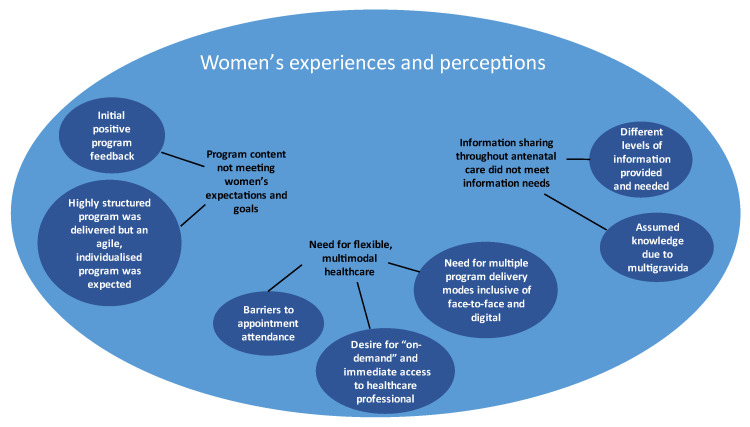
Themes and sub-themes relating to women’s experiences and perceptions of the LWdP program and antenatal care.

**Table 1 nutrients-15-01860-t001:** Characteristics of women consenting to telephone interviews (*n* = 9).

Variable	Median (Range)/Number (%)
Age (years)	31 (22–35)
Highest Level of Education Year 12 or equivalent Associate or undergraduate diploma Bachelor’s degree (including honors) or higher	2 (22) 2 (22) 5 (56)
Average Household Income $50,000–99,999 $100,000–149,999 $150,000 and above	3 (33) 2 (22) 4 (45)
Country of Birth Australia Other English-speaking country Other non-English speaking country	5 (56) 3 (33) 1 (11)
Primary Language Spoken at Home Arabic English Hindi Igbo	1 (11) 6 (67) 1 (11) 1 (11)
Gravida 1 2 3 4	2 (22) 3 (33) 3 (33) 1 (11)
Parity 1 2 3 4	5 (56) 1 (11) 2 (22) 1 (11)
Program Appointments Attended 0 1 2	2 (22) 6 (67) 1 (11)

**Table 2 nutrients-15-01860-t002:** Deductive mapping of interview themes utilizing the BCW, COM-B Model, and TDF Themes.

Themes	Sub-Themes	Theoretical Domains Framework (TDF)	Source of Behavior (COM-B)	Interventions (BCW)	Behavior-Change Techniques	Potential Changes to LWdP and Wider Antenatal Services
1. Program content not meeting women’s expectations and goals	1.1 Initial positive program perceptions	Knowledge (E)	Psychological Capacity	Education	Feedback on the behavior/ outcome(s) of the behavior Biofeedback Self-monitoring of behavior/ outcome of behavior Cue-signaling reward Satiation Prompts/cues Information about antecedents Re-attribution Behavioral experiments Information about social and environmental consequences Information about health consequences Information about emotional consequences Information about others’ approval	Updating LWdP advertising material and welcome letter to include risks of excess GWG and benefits of participating in the LWdP program and having a healthy lifestyle during pregnancy. Expanding the pre-program survey to include: - women’s goals and expectations of the program - current knowledge and skills related to healthy eating, physical activity, and GWG In the first program call: - provide feedback about current eating and exercise behaviors from the pre-program survey and current weight-gain trajectory (if wanted by the patient); - Develop SMART goals based on current behavior feedback, tracking tools, and positive self-talk activities with women; - discuss risks of excess GWG and benefits of healthy lifestyles during pregnancy. Embed self-talk practices into the program. Conduct goal-setting and behavior-change activities with women. Add information about these activities and their benefits to the program manual. Re-visit and collaboratively modify goals if needed with women at the beginning of each phone call. When providing information to women about healthy eating, physical activity, and GWG in pregnancy, always discuss the positives and negatives of their behavior on the pregnancy outcomes and the patient’s and child’s health. Education and training of program dietitians to: - utilize women’s goals, expectations, and existing knowledge and skills to tailor the program progression and information shared; - educate and promote positive self-talk with women. Use social support such as regular peer supervision, including discussing complex cases, use of motivational interviewing, reviewing recorded calls, peer shadowing, etc.
		Optimism (E)	Reflective motivation	Education Modeling	As above AND Use of opinion leaders Feedback Pros and cons Incentive (outcome) Reward (outcome) Self-talk	
	1.2 Highly structured program was delivered but an agile, individualized program was expected	Knowledge (B/E)	Psychological Capacity	Education	As above	
		Skills (cognitive/interpersonal) (B/E)		Training	Instruction on how to perform a behavior Behavioral practice/ rehearsal Graded tasks	
		Beliefs about capabilities (B/E)	Reflective motivation	Enablement	Social support (unspecified) Social support (practical) Social support (emotional) Reduce negative emotions Conserve mental resources Pharmacological support Self-monitoring of behavior/ outcome of behavior Behavior substitution Overcorrection Generalization of a target behavior Graded tasks Avoidance/reducing exposure to cues for the behavior Adding objects to the environment Restructuring the physical environment Restructuring the social environment Distraction Body changes Behavioral experiments Mental rehearsal of successful performance Focus on past success Self-talk Verbal persuasion about capability Self-reward Goal setting (behavior) Goal setting (outcome) Behavioral contract Commitment Action planning Review behavior goal(s) Review outcome goal(s) Discrepancy between current behavior and goal Problem solving Pros and cons Comparative imagining of future outcomes Valued self-identity Framing/reframing Incompatible beliefs Identity associated with changed behavior Identification of self as role model Salience of consequences Monitoring of emotional consequences Anticipated regret Imaginary punishment Imaginary reward Vicarious consequences	
2. Need for flexible, multimodal healthcare	2.1 Barriers to appointment attendance	Social Influences (B)	Social Opportunity	Environmental restructuring	Cue-signaling reward Remove access to the reward Remove aversive stimulus Satiation Exposure Associative learning Reduce prompt/cue Prompts/cue Adding objects to the environment Restructuring the physical environment Restructuring the social environment	Expand the use of digital health to improve program accessibility, e.g., telehealth, email, text messages. Primary points of contact = telephone, telehealth, email. Multiple modes used across a single program. Use digital resources, e.g., videos, websites, downloadable content, digital intake, and weight trackers. Update the promotion of the program to include the multiple available delivery modes and on-demand services. Modify the program and content to be delivered via multiple modes. Train program staff to use alternative program delivery modes, including on-demand services. Modify clinical workloads to allow for the provision of on-demand access to health professionals through email and/or text messages. Use of change champions and social support to upskill, assist, model, and troubleshoot the use of digital technology. Problem solving such as developing “how to” resources for the use of digital technology
	2.2 Desire for “on-demand” and immediate access to healthcare professionals	Environmental context and resources (B/E)	Physical Opportunity	Environmental restructuring Training Enablement	As above	
		Social/professional role and identity	Reflective motivation	Education Modeling Persuasion	As above AND Feedback on the behavior Feedback on the outcome(s) of the behavior Biofeedback Re-attribution Focus on past success Verbal persuasion about capability Persuasive source Framing/reframing Identity associated with changed behavior Identification of self as role model Information about social and environmental consequences Information about health consequences Information about emotional consequences Salience of consequences Information about others’ approval Social comparison	
		Beliefs about capabilities (B/E)				
	2.3 Need for multiple program delivery modes inclusive of face-to-face and digital	Environmental context and resources (B/E)	Physical Opportunity	Environmental restructuring Training	As above	
		Social Influences (B)	Social Opportunity	Environmental restructuring	As above	
		Beliefs about capabilities (B/E)	Reflective motivation	Education Enablement	As above	
3. Information sharing throughout antenatal care did not meet information needs	3.1 Different levels of information provided and needed	Knowledge (B/E)	Psychological Capacity	Education	As above	Ongoing support, education, and training of all HCPs involved in antenatal care to support: - conversations with all patients about healthy eating, exercise, and GWG during pregnancy; - asking patients what information they want about healthy eating, exercise, and GWG during pregnancy; - where and how additional support can be accessed with dedicated resources (physical and digital); Advertisements and information leaflets in maternity clinic rooms and waiting areas that encourage: - discussions about healthy eating, exercise, and GWG in pregnancy; - visual modeling of healthy eating and exercise behaviors in pregnancy; - (self)referral to additional programs and support if needed; - Dietitians and program staff modeling and having discussions with women about healthy eating, exercise, and GWG during pregnancy.
		Environmental context and resources (B)	Physical Opportunity	Training Environmental restructuring Enablement	As above	
		Social/professional role and identity (B/E)	Reflective motivation	Education Training Modeling	As above	
		Beliefs about consequences (E)		Persuasion Modeling	As above	
	3.2 Assumed knowledge due to multigravida	Knowledge (B/E)	Psychological Capacity	Education	As above	
		Social/professional role and identity (B/E)	Reflective motivation	Education Training Modeling	As above	
		Beliefs about consequences (E)		Persuasion Modeling	As above	

## Data Availability

Data are available on request due to privacy and ethical restrictions. The data presented in this study are available on request from the corresponding author. The data are not publicly available to maintain the privacy of the women who participated in the interviews.

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
