# Peer review of "Improving Engagement in Antenatal Health Behavior Programs—Experiences of Women Who Did Not Attend a Healthy Lifestyle Telephone Coaching Program"

_nutrients, 2023, doi:10.3390/nu15081860_

Round 1

Reviewer 1 Report

Thank you for the opportunity to review the manuscript entitled “Understanding the experiences of women referred who did not engage in a nutrition and physical activity telephone coaching program in pregnancy”.

It is common for research on health promotion projects to focus on disseminating the improvements achieved to participants. However, as the authors themselves mention, there is little available literature on the experiences of participating in such programs and on the barriers and enablers to completing them. Thus, the results of this study represent an advance in current knowledge on the aspects to be considered when designing actions aimed at improving the health of pregnant women through diet and physical activity.

Overall, the work is interesting, and the manuscript is presented in a well-structured manner. However, a few parts should be improved to show the information in a clearer way and to be reproducible in other settings. 

The following are some comments that might serve to complete the manuscript.

The word "program" is overused throughout the text of the manuscript (Abstract=6 times; Introduction=12; Methods=5; Results=37; Discussion=31) The use of synonyms or rephrasing is strongly recommended, especially in paragraphs where repetitions are very close.

Title and abstract

It is highly recommended to consider changing the title and improving the content of the abstract in order to make it more attractive to potential readers.

1. Introduction

The background and rationale is adequately explained, although it is suggested to clarify that some of the evidence is drawn from studies conducted in Australia (see lines 58-61) and therefore cannot be generalised worldwide.

2. Materials & Methods

2.1. Study design

Authors are requested to indicate the number of women who were invited to participate (line 112) Was it possible to contact all those who met these criteria? i.e. does the 162 quoted in line 167 match the number of women who met the criteria? This data is crucial to know the response rate of the study.

The text of lines 113 to 116 is included in the introduction and it is suggested to remove it from this paragraph.

2.3. Data analysis

It is very important that this part of the methodology of qualitative studies is clearly described to allow for reproducibility. Therefore, the authors are requested to explain in more detail the paragraph between lines 155 and 161. In particular, it should be clarified which domains of the Theoretical Domains Framework are taken into account (apparently it is the version of Cane, 2012) and what the Behavior Change Wheel/COM-B Model of Behavior Change consist of. It is essential for an easier interpretation of Table 2.

3. Results

Table 1 should show the characteristics of the women participants (line 169). So that, only those of the 9 women who were interviewed (and not of the 10 who consented) should be included. Regarding the format of this table, in order to make the information easier to read, it is recommended to highlight the names of the variables (e.g. "country of birth") with respect to the values that they can take ("Australia" and so on). Besides, it is suggested to omit the Median/range information for the variables “Gravida” and “Parity" and to delete (n(percentage)) next to the latter.

3.1. Part 1: Thematic Analysis

One aspect that is not addressed is whether the information provided by each participant could be related to any of their characteristics. Information on age, number of appointments attended, etc. could be incorporated alongside the assigned number (e.g. Line 193).

3.2. Part 2: Deductive mapping to the TDF, COM-B Model, and BCW

The text should summarise what is indicated in columns 6 and 7 of Table 2 and how this information has been obtained (i.e. criteria were used).

Table 2 is not considered to be entirely self-explanatory (e.g. meaning of letters in parentheses in column 3). As it is laid out, it is difficult to "follow" the information included. Several formatting changes are suggested, especially for the columns with the most information (6 and 7):

- increase the width of the columns

- increase the line spacing between concepts 

- left-aligned

4. Discussion

As a general comment on this section, the topics are adequately discussed. However, the authors, do not relate them to the results of the study itself. The findings of this research need to be referenced along with the related information mentioned in this section.

A very important aspect that needs to be discussed is the sample size of this study. It is lower than recommended for qualitative studies. The response rate is very low (¿around 5%?) which suggests a high risk of bias in the sample and therefore in the results.

5. Conclusions

As a consequence of the above, the conclusions should reflect the location of the study and that they are derived from a sample that does not allow, a priori, for generalisations to be made.

Author Response

Thank you so much for your feedback. Please see below how we have addressed your concerns. 

Reviewer feedback 

How the author has addressed 

Page and line number 

1.1 

The word "program" is overused throughout the text of the manuscript (Abstract=6 times; Introduction=12; Methods=5; Results=37; Discussion=31) The use of synonyms or rephrasing is strongly recommended, especially in paragraphs where repetitions are very close. 

Thank you for your feedback.  

We have gone through and rephrased, removed, or used synonyms for “program” throughout the paper to reduce the number of times it is used. 

Whole paper 

1.2 

Title and abstract 

It is highly recommended to consider changing the title and improving the content of the abstract in order to make it more attractive to potential readers. 

Thank you for your feedback.  

We have updated the title and reworked the abstract. 

Page 1, Lines 2-4, 19-40 

1.3 

Introduction 

The background and rationale is adequately explained, although it is suggested to clarify that some of the evidence is drawn from studies conducted in Australia (see lines 58-61) and therefore cannot be generalised worldwide. 

Thank you for your feedback.  

We have added additional references to clinical practice guidelines in the UK and USA supporting dietitian referral for women with pre-pregnancy BMIs >25kg/m2. 

We have clarified barriers to care related are specific to the Australian antenatal setting based on the evidence we have referenced. 

We have also added a statement in our discussion addressing lack of global generalizability of results based on investigated cohort. 

Page 2, Line 46 

Page 2, Line 49-50 

Page 17, Lines 381-382 

1.4 

Materials & Methods 

2.1. Study design 

Authors are requested to indicate the number of women who were invited to participate (line 112) 

Thank you. Number of women contacted has been clarified in results.  

Page 4, Line 157 

1.5 

Was it possible to contact all those who met these criteria? i.e. does the 162 quoted in line 167 match the number of women who met the criteria? This data is crucial to know the response rate of the study. 

Addressed in Point 1.4 

1.6 

The text of lines 113 to 116 is included in the introduction and it is suggested to remove it from this paragraph. 

Thank you.  

Previous lines 113-116 have been removed as recommended. 

Page 3, Lines 117-118 

1.7 

2.3. Data analysis 

It is very important that this part of the methodology of qualitative studies is clearly described to allow for reproducibility. Therefore, the authors are requested to explain in more detail the paragraph between lines 155 and 161. In particular, it should be clarified which domains of the Theoretical Domains Framework are taken into account (apparently it is the version of Cane, 2012) and what the Behavior Change Wheel/COM-B Model of Behavior Change consist of. It is essential for an easier interpretation of Table 2. 

Thank you for your feedback. 

We have added clarifying statements on which TDF domains were used and have added an additional picture to demonstrate the BCW/COM-B Model components.  

Page 3, Lines 142-145 

Page 4, Figure 1, Lines 154-155 

1.8 

3. Results 

Table 1 should show the characteristics of the women participants (line 169). So that, only those of the 9 women who were interviewed (and not of the 10 who consented) should be included. Regarding the format of this table, in order to make the information easier to read, it is recommended to highlight the names of the variables (e.g. "country of birth") with respect to the values that they can take ("Australia" and so on). Besides, it is suggested to omit the Median/range information for the variables “Gravida” and “Parity" and to delete (n(percentage)) next to the latter. 

Thank you. Table has been reformatted to only include information of the nine women interviewed. Have made suggested edits to highlight key variables, remove median for gravida and parity, and remove (n(percentage)). 

Page 4, Line 162 (Table 1) 

1.9 

3.1. Part 1: Thematic Analysis 

One aspect that is not addressed is whether the information provided by each participant could be related to any of their characteristics. Information on age, number of appointments attended, etc. could be incorporated alongside the assigned number (e.g. Line 193). 

Thank you.  

We have incorporated women’s age and appointments attended as suggested next to each quote throughout the paper. 

Pages 6-8, Lines 178-181, 185-186, 195-197, 206-207, 215-216, 220-221, 226-228, 233-234, 241-243, 246-247, 256-259, 266-267, 273-275, 279-280, 287-288, 292-293 

1.10 

3.2. Part 2: Deductive mapping to the TDF, COM-B Model, and BCW 

The text should summarise what is indicated in columns 6 and 7 of Table 2 and how this information has been obtained (i.e. criteria were used). 

Table 2 is not considered to be entirely self-explanatory (e.g. meaning of letters in parentheses in column 3). As it is laid out, it is difficult to "follow" the information included. Several formatting changes are suggested, especially for the columns with the most information (6 and 7): 

- increase the width of the columns 

- increase the line spacing between concepts  

- left-aligned 

Thank you for your feedback.  

We did find formatting this table especially difficult. We have implemented your suggested formatting changes and added further detail and clarification prior to the table as suggested.  

Pages 9-15, Table 2 

Page 8, Lines 302-308 

1.11 

4. Discussion 

As a general comment on this section, the topics are adequately discussed. However, the authors, do not relate them to the results of the study itself. The findings of this research need to be referenced along with the related information mentioned in this section. 

A very important aspect that needs to be discussed is the sample size of this study. It is lower than recommended for qualitative studies. The response rate is very low (¿around 5%?) which suggests a high risk of bias in the sample and therefore in the results. 

Thank you for your feedback. 

We have reworked discussion points to clearly highlight and incorporate the results of our study. 

We have also added additional discussions to the limitations of our study including sample size and potential lack of generalizability of results.  

Pages 16-17, Lines 312-404 

Page 16, lines 390-404 

1.12 

5. Conclusions 

As a consequence of the above, the conclusions should reflect the location of the study and that they are derived from a sample that does not allow, a priori, for generalisations to be made. 

Thank you for your feedback. 

We have reworded our conclusion to more directly relate results to our local site and program. 

Page 17, Lines 406-145 

Reviewer 2 Report

This study described the barriers faced by 10 women to engage in an antenatal healthy eating and physical activity program. Overall, this is a well-written manuscript. A few suggestions for improvement below:

1. Of 162 women, only 9 was interviewed. Can the authors elaborate on the characteristics of women who did not consent? This will provide greater perspective on the hard-to-reach group. 

2. Can the authors elaborate on why TDF, BCW/COM-B were selected, out of the many behaviour change theories? 

3. Is the interviewer trained in conducting the interviews and thematic analysis? 

Author Response

Thank you for your feedback. Please see how we have addressed your concerns below.

Reviewer feedback 

How the author has addressed 

Page and line number 

2.1 

Of 162 women, only 9 was interviewed. Can the authors elaborate on the characteristics of women who did not consent? This will provide greater perspective on the hard-to-reach group.   

Thank you for your feedback. 

We have added a supplementary table with the non-responder's characteristics. 

Table S1 

2.2 

Can the authors elaborate on why TDF, BCW/COM-B were selected, out of the many behaviour change theories?   

Thank you for your feedback.  

We have clarified our selection of the TDF, BCW/COM-B Models in this study. 

*Need to add in reference – see Shelley’s email* 

Page 3, Lines 137-141 

2.3 

Is the interviewer trained in conducting the interviews and thematic analysis?   

Thank you. 

We have added a clarifying statement on the interviewer's credentials. 

Page 3, Lines 119-120